# High-Performanced Hemicellulose Based Organic-Inorganic Films with Polyethyleneimine

**DOI:** 10.3390/polym13213777

**Published:** 2021-10-31

**Authors:** Han Wu, Jing Li, Yule Wu, Hui Gao, Ying Guan

**Affiliations:** 1School of Forestry & Landscape Architecture, Anhui Agricultural University, Hefei 230036, China; Wuhanahau@163.com (H.W.); lijingaau@163.com (J.L.); m18855191896@163.com (Y.W.); 2Biomass Molecular Engineering Center, Anhui Agricultural University, Heifei 230036, China

**Keywords:** quaternized hemicelluloses, bentonite, polyethyleneimine, composite films, mechanical property

## Abstract

For the high-value utilization of hemicellulose-based composite films, the poor film-forming and mechanical properties of hemicellulose-based composite films must be surmounted crucially. Based on this, hemicellulose-based organic-inorganic composite films with good mechanical properties were prepared from quaternized hemicelluloses (QH), bentonite, and polyethyleneimine (PEI). The QH/PEI/bentonite composite films were prepared by vacuum filtration, and the properties of the composite film were investigated. The results showed that the QH was inserted into bentonite nanosheets through hydrogen bonding and electrostatic interactions. PEI was cross-linked with hemicellulose by hydroxyl groups, electrostatically attracted by the bentonite flake layers. The mechanical properties of the composite films were significantly increased by the incorporation of PEI. When the PEI content was 20%, the tensile stress of the composite film was increased by 155.18%, and the maximum tensile stress was reached 80.52 MPa. The composite films had strong UV absorption ability with the transmittance was almost 0 in the UV region from 200 to 300 nm. The thermal property of composite film was also improved, and the residual mass increased by three times compared to QH. These results provide a theoretical basis for the use of hemicellulose-based composite films in packaging applications.

## 1. Introduction

In recent years, there had been growing interest in the development of materials from natural polymers, and those obtained from renewable resources particularly [1,2]. Among natural polymers, ligno-cellulosic materials are potentially valuable resources because of the conversion into biofuels and bio-products [3,4]. Hemicelluloses, the most abundant biomass resources, which are second only to cellulose [5], have received ever increasing interest due to lower cost, oxygen barrier properties, and easy availability from agriculture [6]. Hemicellulose is a highly branched heteropolymer of hexose (e.g., mannose, glucose, galactose), pentose (e.g., xylose, arabinose), and uronic acid (e.g., glucuronic) sugar residues [7]. As an inexpensive, biodegradable and renewable resource, hemicellulose has many advantages such as good biocompatibility, oxygen barrier and optical transparency, which make it useful for biodegradable films [8]. However, hemicelluloses are hydrophilic in nature, and the hemicellulose-based films are proved to be semi-crystalline, hygroscopic, and of poor mechanical strength [9]. Hemicelluloses have an abundance of free hydroxyl groups, an ideal candidate for chemical modification [10], such as carboxymethyl [11], acylated [12], and cationic hemicelluloses [13]. Chemically modified hemicelluloses could be used to prepare materials with unique properties which could improve the value of the biopolymers [14].

It is also well known that the solubility and yields of hemicelluloses can be enhanced by the quaternization of hemicelluloses [15,16], and novel opportunities to maximally exploit the various valuable properties of hemicelluloses for previously unperceived applications would be created [17,18]. Rao et al. reported that quaternized hemicellulose-based films exhibit excellent mechanical properties and moisture sensitivity. The composite films showed promising applications in biomedicine, packaging materials, humidity sensors and other fields [19]. In addition, the additive is often needed to ensure mechanical properties of hemicelluloses films, such as plasticizer and filler. Optimization of this structure can be achieved by adding clay or plasticizer [20]. Clay is an important nanomaterial for biopolymer modification [21]. It is reported that the biomaterial obtained with clay added to biopolymers has higher thermal stability, mechanical strength, and better barrier properties [22,23,24]. Guan et al. studied organic-inorganic composite films based on quaternized hemicelluloses (QH) and clay nanosheets. The clay nanosheets endowed the hemicellulose based composite films with good thermal stability and UV Vis transparency [25]. Bentonite is aluminum phyllosilicate clay, which is presented in the form of platelets, formed by two layers of tetrahedral silicate with an octahedral blade in the center of each layer and exchangeable cations between the layers [26,27]. The cations between the layers are easily exchanged and have a large ion exchange capacity. Thus, bentonite has strong adsorption capacity, good hygroscopic swelling and strong ion exchange capacity [28].

To improve the performance of quaternized hemicelluloses/bentonite hybrid materials, some additives can be introduced as fillers to enhance the mechanical behavior. Chen studied the reinforcement of quaternized hemicellulose/montmorillonite composite films by two fillers, polyvinyl alcohol (PVA) and chitin nanowhiskers (NCH). It was found that the mechanical strength, thermal stability, transparency of hemicellulose based composite films can be well improved [29]. Polyethylenimine (PEI) has high adhesion and adsorption with the structure of polar group (amino) and hydrophobic group (vinyl) and can combine with different substances. Polyethyleneimine exists in water as a polymer cation, which can neutralize and adsorb all anionic substances. Due to the high cationic degree, it can promote the close combination of hemicellulose and bentonite, thus forming a more stable composite film structure. In the current study, quaternized hemicelluloses (QH) were used as raw material to prepare hemicellulose/PEI/bentonite composite films by intercalation of quaternized hemicelluloses and bentonite with the addition of different proportions of polyethyleneimine. The structure, morphology, optical transparency and thermal stability of the composite films were analyzed by fourier transform infrared spectrometer (FT-IR), X-ray diffraction (XRD), scanning electron microscope (SEM), thermogravimetric analysis (TGA), transparency and tensile properties. These results might help to interpret the relationship between structures and properties of the composite films and explore the potential applications of composite films.

## 2. Materials and Methods

### 2.1. Materials

The QH was obtained from the previous work [30], which average molecular weight (Mw) was 9240 g mol^−1^ tested by size exclusion chromatography (SEC). The bentonite was purchased from the macro bentonite factory in xingyangpingqiao District. The PEI was purchased from the Shanghai Aladdin Bio-Chem Technology Co., Ltd., Shanghai, China, which average molecular weight (Mw) was 1800 g mol^−1^. Filter membrane was purchased from Jinteng, Tianjin, China (polyvinylidene fluoride microfiltration membrane, 0.45 μm average pore diameter). 2,3-Epoxypropyltrimethylammonium chloride (ETA) was purchased from Sigma-Aldrich Co., Saint Louis, MO, USA. The reagents used in this study were all analytical reagents.

### 2.2. Preparation of Organic-Inorganic Film

5 g bentonite powder was dissolved in 500 mL deionized water to be configured into a 1 wt% suspension. 1 wt% bentonite suspension was prepared by stirring at 1000 rpm for 30 min, and then expanded and exfoliated by an ultrasonic processor from Scientz-II D (Ningbo Scientz Biotechnology Co., Ltd., Ningbo, China). This process was repeated three times, followed by centrifugation of the solution at 3800 rpm for10 min. The obtained bentonite suspension was used for composite preparation. The concentrations of QH and PEI were all kept at 1wt%. Hemicellulose-based organic-inorganic films with polyethyleneimine were prepared by vacuum suction filtration. Firstly, The QH solution was mixed with the obtained bentonite suspension, and the volume ratios of the two substances were kept at 1/1, obtaining the QH- bentonite matrix. Secondly, different contents of PEI were added (0 wt%, 5 wt%, 10 wt%, 15 wt%, 20 wt%) to the QH- bentonite blend liquid. All the mixed solution volume was at 15 mL, and then was magnetically stirred for 24 h at 30 °C. Then, the QH bentonite blend liquid was obtained after vacuum filtration of the filter membrane for 20min. Finally, the composite films were vacuum dried for 15 min at 80 °C. The film obtained without Pei was designated as PEI 0%. The composite films with different contents of PEI (5 wt%, 10 wt%, 15 wt%, 20 wt%) were named as PEI 5%, PEI 10%, PEI 15%, and PEI 20%, respectively.

### 2.3. Characterization of the Films

The FT-IR spectra of the QH, bentonite and composite films were measured on a Thermo Scientific Nicolet In 10 FT-IR Microscope (Thermo Nicolet Corporation, Madison, WI, USA) equipped with a liquid nitrogen cooled MCT detector. Dried samples were recorded with BaF_2_ disks in the range from 4000 to 650 cm^−1^ at a resolution of 4 cm^−1^ and 128 scans per sample. The crystallinities of the QH, QH, bentonite and the composite films were measured using an XRD-6000 instrument (Shimadzu, Japan) in reflection mode in the angular range of 5–40° (2θ) at a speed of 5°/min. The measurement was carried out with a Cu Kα radiation source (λ = 0.154 nm) at 40 kV and current 35 mA. Thermal behavior of the three films was performed using thermogravimetric analysis (TGA) and derivative thermogravimetry (DTG) on a simultaneous thermal analyzer (DTG-60, Shimadzu) under a nitrogen atmosphere from 25 to 700 °C and with a heating rate of 20 °C/min. The morphology of film samples was investigated by field emission scanning electron microscopy using a Hitachi S-3400N II (Hitachi, Japan) instrument at 15 kV. 

### 2.4. The Mechanical Properties of the Films

The tensile tests of the films were performed with a universal materials testing machine (UTM6503, Shenzhen Suns Technology Stock Co. Ltd., Shenzhen, China). Specimens of 20 mm length, 20–30 um thickness, 10 mm width were tested with strain rate of 5 mm/min. The relative humidity was kept at 50% under the room temperature. The mechanical tensile data for each sample measured three repeated and the average was used to determine the mechanical properties.

### 2.5. UV-Vis Transparency of the Composite Films 

The UV-vis spectra of sample films were collected by an ultraviolet/visible spectrophotometer (Beijing Purkinje General Instrument Co., Ltd., Beijing, China) in the range of 200–800 nm. Before UV-vis measurements, the films were pasted on the surface of quartz pool.

## 3. Results

### 3.1. Structure of the Composite Films

The films prepared from hemicellulose were always cracks and defects due to the strong internal stress by hydrogen bondings [31]. However, the hemicellulose-based films could be formed by hydrogen bonds between the hemicellulose and bentonite. The mechanical strength of the film from biomacromolecular was mainly determined by hydrogen bonding, and the reduction of hydrogen bonding always led to the decrease of the mechanical strength [32]. In this study, when the bentonite suspended in water, the bentonite flake layer was peeled off, forming a negatively charged surface [33]. In addition, the positive charges were in the QH solution. Then, the electrostatic attraction between two oppositely charged polymers was the driving force for the formation of multilayered films. The multilayer structure could be formed attributed to the electrostatic interactions and hydrogen bonds between the bentonite and QH [34,35]. PEI, as the reinforcement, could be homogeneously dispersed in the hemicellulose and bentonite matrices. The surface roughness of the film could be increased by the presence of bentonite, of which might play an important role in thermal behavior. For the formation of the high- performance film, the hemicellulose was embedded with the bentonite flakes by hydroxyl groups and electrostatical attraction, and PEI was cross-linked with hemicellulose by hydroxyl groups, facilitating the binding of QH to the bentonite flake layer. The mechanical strength of the composite films might be improved due to the more compact structure (Figure 1).

The FT-IR spectra of QH, bentonite, and the composite films were shown in Figure 1A,B. From the QH spectrum, the characteristic peaks at 3357 cm^−1^, 2923 cm^−1^, 1615 cm^−1^, 1046 cm^−1^ and 896 cm^−1^ were originated from the stretching vibrations of –OH, C–H, C=O in glyoxalate and β(1-4)-glycosidic bonds among the chains of hemicellulose molecular, respectively [19,36]. The characteristic peak at 1474 cm^−1^ was attributed to the stretching vibrations of methyl and methylene groups on quaternary ammonium substituents [37]. A characteristic absorption peak at 1047 cm^−1^ was due to Si–O stretching vibrations in bentonite [25]. The peak was observed at 797cm^−1^ was assigned for Al–OH bending vibrations [38]. The spectra of the composite films were shown in Figure 1B. The characteristic peaks at 3345 cm^−1^ and 2900 cm^−1^ were assigned to hydroxyl and methylene group contraction vibrations, respectively. The weak peak at 1449 cm^−1^ was assigned to C–H stretching of PEI [39]. A signal at 995 cm^−1^ originated from the vibration of bentonite, indicating that the Si−QH bonds were formed through the OH group of the QH interacting with Si on the bentonite surface [40]. No new characteristic peaks were presented in the spectrum of composite film, indicating that QH was intercalated into the bentonite nanoplatelets with no chemical cross-linking occurred among the components of the composite films. Due to the strong electrostatic and hydrogen bonding effects, the molecular chains were aggregated or rearranged during moisture removal. The spatial positions among molecules produces were changed, and then cross-linked structure was formed between bentonite and QH [15].

X-ray diffraction analysis was carried out to determine the dispersion of the bentonite layers in the matrix of the hemicelluloses. In Figure 1E, the characteristic diffraction peaks at 2θ = 7.6°, 27.9° were attributed to bentonite [41], while the characteristic diffraction peaks at 2θ = 23.1°was attributed to QH [42]. The XRD patterns of the five composite films were shown in Figure 1C,D. The composite film had a certain crystal structure, and the diffraction peaks appeared at 19.4° and 23.1°, and characteristic peak at 2θ = 7.6° was attributed to bentonite. The interplanar spacing of the PEI 0%, PEI 5%, PEI 10%, PEI 15%, PEI 20% calculated by Bragg’s equation were 1.16 nm, 1.06 nm, 1.08 nm, 1.13 nm, 1.13 nm, respectively. By comparing the interplanar spacing, the interplanar spacing of the composite film was smaller after the addition of PEI. After the addition of PEI, QH could be combined more effectively with bentonite, which was more tightly packed with ordered structured layered structure formed by self-assembly, and the mechanical properties of the composite films would be improved. This result will be verified in the mechanical analysis.

### 3.2. Morphology of the Composite Films

The images of the surface and cross section of the PEI 0% (A_1_, A_2_) and PEI 10% (B_1_, B_2_) were presented in Figure 2. As shown in A_1_ and B_1_, the surface of the film was smooth and homogeneous, which indicated that the patterns of the composite films were improved by the introduction of bentonite nanoplatelets during the electrostatic interaction. In the cross section of A_2_ and B_2_, the lamellar structure of the composite film was clearly observed, which was due to the original structure of bentonite nanoplatelets. The orientation of bentonite nanoplatelets in the film was possibly caused by directional flow induced by vacuum filtration and the electrostatic interaction and hydrogen bonding. The electrostatic attractions between two opposite charged polymers were the driving force for the formation of composite films. During the electrostatic interaction, the composite films were deposited by an alternating sequence of negatively and positively charged layers. The close bonding of QH with bentonite was probably promoted by PEI, which would be enabled the formation of a stable composite film structure. That was an important reason for the improved mechanical properties of the composite film.

### 3.3. Mechanical Properties of the Composite Films

The effects of different contents of PEI on the mechanical properties of the composite films were investigated in Figure 3, and the data was presented in Table 1. As can be seen from Figure 3 and Table 1, the tensile strength and Young’s modulus of the composite films were increased with an increase in PEI content, following the order PEI 20% > PEI 15% > PEI 10% > PEI 5% > PEI 0%. The tensile strength of composite film was 22.67 MPa prepared without the PEI, which was higher than that of the composite film (19.8 MPa) prepared from QH and montmorillonite according to the study of Chen [29]. This might be due to the bentonite used in this study was easier to dissolve in the QH matrix. The tensile strengths of the composite films were improved with increasing of PEI content. When the PEI content was 20%, the maximum tensile strength of the composite film was reached to 80.52 MPa with the Young’s modulus of 8.14 GPa, which was almost fourfold increase compared with PEI 0%. This might be due to that the QH and bentonite could be combined more compact with the addition of PEI. However, the composite films exhibited a substantial rough surface as a result of the tightly connected of bentonite nanoplatelets and PEI with the QH macromolecular chain. The quaternized hemicellulose was more tightly bound to the bentonite and the interlayer more tightly packed with the addition of PEI. The flexible QH chains and the rigid bentonite flakes were complexed more fully as the content of PEI increased. Therefore, the mechanical properties of the composite films were effectively improved.

### 3.4. UV-Vis Transparency of the Composite Films

UV-Vis light transmittances of quaternized hemicelluloses/bentonite/PEI composite films were shown in Figure 4. The transparencies of the five composite films were all gradually increased from the UV light to visible light. In addition, the transmittances of the composite films with different QH contents were almost 0 in the UV region from 200 to 300 nm, which showed that the composite films had strong UV absorption ability. This phenomenon indicated that composite films could be used for applications in packaging and coating. It was worth pointing out that the light transmittance of composite film was influenced by the content of PEI to a certain extent (as shown in Figure 5). When PEI contend was 10%, the light transmittance of the composite film was up to 88%, however, the light transmittance of the composite film was decreased after PEI content was more than 10%. It might be that the formation of hydrogen bonds between bentonite nanoplatelet and QH molecular chains was promoted by PEI. The visible light transmittance of the composite films was affected under the hydrogen bonding.

### 3.5. Thermal Properties of the Composite Films

The thermal stabilities of the QH, bentonite, and composite films were analyzed by thermogravimetric (TG), and the results were shown in Figure 6. The maximum rates of decomposition and weight loss of the composite films were obtained at temperatures from 200 °C to 600 °C Further, the characteristic temperature (Tonset is the temperature at the onset of decomposition, T1 is the maximum weight loss temperatures) of the samples were shown in Table 2. The weight loss below 100 °C was mainly due to the moisture release of the samples. The remaining mass of QH was 17.3% at 700 °C, indicating that the thermal stability of QH was poor with the vigorously pyrolyzed molecular chains at high temperature. Therefore, the thermal stability of QH should be improved by adding other components such as bentonite. From the TGA curve of bentonite in Figure 5A, there was still 95.3% residue mass of bentonite when the temperature reached 700 °C, which indicated that bentonite had a relatively high thermal stability. The unique octahedral structure confers good thermal resistance to bentonite, which is lost a small part of its weight. This has been well demonstrated in studies by Costafreda and Alther [43,44]. The residual weights of PEI 0% and PEI 10% at 700 °C were 57.3%, and 56.5%, respectively, which were considered that the major weight loss of the composite films was caused by the loss of cations from the QH. The T_onset_ values of QH, PEI 0% and PEI 10% were obtained at 119.6 °C, 184.7 °C, and 220.7 °C, respectively. Furthermore, the T_onse_ of the composite films was increased markedly relative to the bentonite. This reduction was attributed to the high thermal stability of the bentonite. Consequently, as the content of PEI increased, the thermal stability of the composite film was increased, which was due to that the composite film would be exhibited high performance by adding the PEI.

## 4. Conclusions

Quaternized hemicelluloses (QH) were used as raw material to prepare hemicellulose/PEI/bentonite composite films by intercalation of quaternized hemicelluloses and bentonite by adding different proportions of polyethyleneimine with vacuum filtration.This result suggested that no chemical reaction was occurred between bentonite and QH, and QH was intercalated into the bentonite nanoplatelets. The bentonite nanoplatelets were uniformly dispersed in the QH matrix, despite with the attendance of PEI. The layered structure of composite film was obtained, and the surfaces of the composite films were homogeneous and smooth. The mechanical properties of the composite films were effectively improved by the addition of PEI. The tensile strengths of the composite films were improved with increase of PEI content. When the PEI content was 20%, the maximum tensile strengths of the composite films were reached to 80.52 MPa. Moreover, the thermal stability properties of the composite films were effectively improved by PEI. These properties indicated that the performance of the composite films could be effectively improved by PEI. In addition, composite films have thermal stability and UV resistance. These characteristics provide a theoretical basis for packaging applications of hemicellulose based composite films.

## Data Availability

The data presented in this study are available upon request from the corresponding author.

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
