# Peer review of "High-Performanced Hemicellulose Based Organic-Inorganic Films with Polyethyleneimine"

_polymers, 2021, doi:10.3390/polym13213777_

Round 1

Reviewer 1 Report

This manuscript presents a study about properties of hemicellulose based organic-inorganic films with polyethyleneimine. The work has some potential. However, some points listed below need to be improved. I suggest major revision.  

Introduction section: please correct the typo, it is “ligno” instead of “lingo”.

Introduction section: clearer the novelty of this work in the introduction section. I also suggest add some works related to the usage of hemicellulose to developed polymeric films.

Section 2.1: What are the main properties of  polyethyleneimine?   

Section 2.2 (first sentence): How many bentonite and water were used to prepare the suspension? What is the conditions used in the ultrasonification. How many hours the suspension was ultrasonificated?

Section 2.2: Are the different contents of PEI used added in mass (wt%)?

Section 2.2: better describe how PEI was added to the mixture. In addition, describe how the films were obtained. What method was used?

Section 2.4: how many specimens were tested?

Figure 1: please improve the quality of this figure. It is very hard to see the bands from FTIR.

Section 3.2: I suggest add the micrographs of PEI 0% for better compare the results.

Section 3.3 and Table 1: the authors must add the standard deviation for all values presented in Table 1 and discussed in section 3.3. What is the film thickness?

Section 3.4: I suggest add more images of the film developed.

Section 3.5: Why the authors did not present the TGA results from PEI 5%, PEI 15% and PEI 20% samples?

Author Response

  1. Introduction section: please correct the typo, it is “ligno” instead of “lingo”.

Replying:

The “lingo” has been revised into “ligno”, and marked in red.

  1. Introduction section: clearer the novelty of this work in the introduction section. I also suggest add some works related to the usage of hemicellulose to developed polymeric films.

Replying:

The contents “Hemicellulose is a highly branched heteropolymer of hexose (e.g., mannose, glucose, galactose), pentose (e.g., xylose, arabinose), and uronic acid (e.g., glucuronic) sugar residues. As an inexpensive, biodegradable and renewable resource, hemicellulose has many advantages such as good biocompatibility, oxygen barrier and optical transparency, which make it useful for biodegradable films.” and “Recently, Rao et al. reported that quaternized hemicellulose based films exhibit excellent mechanical properties and moisture sensitivity. The composite films showed promising applications in biomedicine, packaging materials, humidity sensors and other fields.” have been added in the Introduction Part, and marked in red.

  1. Section 2.1: What are the main properties of polyethyleneimine?

Replying:

The contents “The PEI was purchased from the Shanghai Aladdin Bio-Chem Technology Co., LTD., which average molecular weight (Mw) was 1800 g mol-1.” have been added in Section 2.1, and marked in red.

  1. Section 2.2 (first sentence): How many bentonite and water were used to prepare the suspension? What is the conditions used in the ultrasonification. How many hours the suspension was ultrasonificated?

Replying:

The “A certain amount of bentonite powder was dispersed in deionized water by ultrasonication to form a stable suspension. 1 wt % bentonite suspension was prepared by stirring at 1000 rpm for 30 min and then exfoliating by an ultrasonic processor from Scientz-II D(Ningbo Scientz Biotechnology Co., Ltd). This process was repeated three times, followed by centrifugation of the solution at 3800 rpm for 10 min.” has been revised into “5 g bentonite powder was dissolved in 500 mL deionized water to be configured into a 1 wt% suspension. 1 wt % bentonite suspension was prepared by stirring at 1000 rpm for 30 min, and then expanded and exfoliated by an ultrasonic processor from Scientz-II D (Ningbo Scientz Biotechnology Co., Ltd). This process was repeated three times, followed by centrifugation of the solution at 3800 rpm for10 min.”, and marked in red.

  1. Section 2.2: Are the different contents of PEI used added in mass (wt%)?

Replying:

Yes, The contents “A series of composite films were obtained by adding different contents of PEI (5%, 10%, 15%, 20%) with the same steps as the PEI 0%.” has been revised into “Hemicellulose-based organic-inorganic films with polyethyleneimine were prepared by vacuum suction filtration. Firstly, The QH solution was mixed with the obtained bentonite suspension, and the volume ratios of the two substances were kept at 1/1, obtaining the QH-bentonite matrix. Secondly, different contents of PEI were added (0 wt%, 5wt%, 10wt%, 15wt%, 20wt%) to the QH- bentonite blend liquid. All the mixed solution volume was at 15 mL, and then was magnetically stirred for 24 h at 30 oC.”, and marked in red.

  1. Section 2.2: better describe how PEI was added to the mixture. In addition, describe how the films were obtained. What method was used?

Replying:

The contents “A certain amount of bentonite powder was dispersed in deionized water by ultrasonication to form a stable suspension. 1 wt % bentonite suspension was prepared by stirring at 1000 rpm for 30 min and then exfoliating by an ultrasonic processor from Scientz-II D(Ningbo Scientz Biotechnology Co., Ltd). This process was repeated three times, followed by centrifugation of the solution at 3800 rpm for10 min. The 1 wt% QH solution was mixed with the obtained 1 wt % bentonite suspension, and the volume ratios of the two substances were kept at 1/1. The mixed solution with a total volume of 15 mL was stirred at 30℃for 24 h and then was vacuum filtrated with the filter membrane for 20 min. The obtained film was named as PEI 0%. A series of composite films were obtained by adding different contents of PEI (5%, 10%, 15%, 20%) with the same steps as the PEI 0%. The films obtained with the PEI were named as PEI 5%, PEI 10%, PEI 15%, and PEI 20%, respectively.” have been revised into “5 g bentonite powder was dissolved in 500 mL deionized water to be configured into a 1 wt% suspension. 1 wt% bentonite suspension was prepared by stirring at 1000 rpm for 30 min, and then expanded and exfoliated by an ultrasonic processor from Scientz-II D (Ningbo Scientz Biotechnology Co., Ltd). This process was repeated three times, followed by centrifugation of the solution at 3800 rpm for10 min. The obtained bentonite suspension was used for composite preparation. The concentrations of QH and PEI were all kept at 1wt%. Hemicellulose-based organic-inorganic films with polyethyleneimine were pre-pared by vacuum suction filtration. Firstly, The QH solution was mixed with the obtained bentonite suspension, and the volume ratios of the two substances were kept at 1/1, obtaining the QH- bentonite matrix. Secondly, different contents of PEI were added (0 wt%, 5wt%, 10wt%, 15wt%, 20wt%) to the QH- bentonite blend liquid. All the mixed solution volume was at 15 mL, and then was magnetically stirred for 24 h at 30 oC. Then, the QH bentonite blend liquid was obtained after vacuum filtration of the filter membrane for 20min. Finally the composite films were vacuum dried for 15min at 80 oC. The film obtained without Pei was designated as PEI 0%. The composite films with different contents of PEI (5wt%, 10wt%, 15wt%, 20wt%) were named as PEI 5%, PEI 10%, PEI 15%, and PEI 20%, respectively.”, and marked in red.

  1. Section 2.4: how many specimens were tested?

Replying:

The sentence “The mechanical tensile data for each sample measured three repeated and the average was used to determine the mechanical properties.” has been added in Section 2.4.

  1. Figure 1: please improve the quality of this figure. It is very hard to see the bands from FTIR.

Replying:

The Figure 1 has been improved as follows.

Figure 1.

  1. Section 3.2: I suggest add the micrographs of PEI 0% for better compare the results.

Replying:

The SEM images of PEI 0% have been added in Figure 2.

Figure 2. SEM images of the surface (A1, B1) and cross section (A2, B2) of the composite films.

  1. Section 3.3 and Table 1: the authors must add the standard deviation for all values presented in Table 1 and discussed in section 3.3. What is the film thickness?

 Replying:

Yes, the standard deviation has been added in Table 1.

Table 1. The results of tensile tests of the composite films.

Sample

Thickness (um)

Tensile strength

(MPa)

Tensile strain at fracture(%)

Young’s modulus

(GPa)

PEI 0%

71±0.87

22.67±0.65

0.92±0.22

2.41±0.29

PEI 5%

73±0.93

25.94±0.71

0.51±0.12

4.31±0.59

PEI 10%

76±0.62

28.96±0.42

0.81±0.31

6.08±0.13

PEI 15%

78±0.53

45.54±0.53

0.70±0.31

7.15±0.17

PEI 20%

80±0.24

80.52±0.33

1.30±0.84

8.14±0.04

  1. Section 3.4: I suggest add more images of the film developed.

Replying:

The films were all fabricated from the same component with different ratios. Therefore, the shapes and images of the films were little difference. The only image presented in the text was just for typical examples.

  1. Section 3.5: Why the authors did not present the TGA results from PEI 5%, PEI 15% and PEI 20% samples?

Replying:

The mechanism of bentonite in QH matrix was the mainly research in this study, and the TGA were mainly analyzed the film from QH and bentonite. The function of PEI was mainly at tensile and optical properties. Therefore, the PEI 0% and PEI 10% were presented for examples.

Reviewer 2 Report

Manuscript entitled “High-Performanced Hemicellulose Based Organic-inorganic Films with Polyethyleneimine” could be interesting for the readers. However, the paper needs a major revision before publication. I have listed a few comments that need to be addressed:

  1. Introduction is short, it could be much better with more background about the work with up-to-date citations. Also, I would advise to add few recent reviews on this topic in the introduction part.
  2. What is the novelty of this work that should be clearly addressed at the end of the introduction?
  3. The film fabrication method is not clear? Add more details.
  4. Write the full form once when mentioning for the first instance.
  5. Improve the quality of all the Figures.
  6. Add apparent images of all the fabricated films.
  7. Why is only one SEM image is showed in Fig. 2? Add SEM images of all the fabricated composite films.
  8. Why the residual char content is very high, need explanation?
  9. What about the hydrodynamic properties such barrier, hydrophobicity and water solubility of the film? Add this information.
  10. Add statistical analysis with significant differences in Table 1 & 2.
  11. Authors are advised to add more discussion of the obtained results with up-to-date citations.
  12. Also, carefully revise the typos and linguistic errors to make the manuscript error-free.

Author Response

  1. Introduction is short, it could be much better with more background about the work with up-to-date citations. Also, I would advise to add few recent reviews on this topic in the introduction part.

Replying:

  The contents “Hemicellulose is a highly branched heteropolymer of hexose (e.g., mannose, glucose, ga-lactose), pentose (e.g., xylose, arabinose), and uronic acid (e.g., glucuronic) sugar residues. As an inexpensive, biodegradable and renewable resource, hemicellulose has many advantages such as good biocompatibility, oxygen barrier and optical transparency, which make it useful for biodegradable films.”, “Recently, Rao et al. reported that quaternized hemicellulose based films exhibit excellent mechanical properties and moisture sensitivity. The composite films show promising ap-plications in biomedicine, packaging materials, humidity sensors and other fields.”, “Guan et al. Studied organic-inorganic composite films based on quaternized hemicelluloses (QH) and clay nanosheets. The clay nanosheets endowed the hemicellulose based composite films with good thermal stability and UV Vis transparency”, and “Chen studied the reinforcement of quaternized hemicellulose/ montmorillonite composite films by two fillers, polyvinyl alcohol (PVA) and chitin nanowhiskers (NCH). It was found that the mechanical strength, thermal stability, transparency of hemicellulose based composite films can be well improved.” have been added in the Introduction Part, and marked in red.

  1. What is the novelty of this work that should be clearly addressed at the end of the introduction?

Replying:

The films prepared from hemicelluloses were always brittle, easy to absorb moisture, difficult to form film, etc. However, the thermal and mechanical properties could be improved by adding the bentonite nanosheets. And the bentonite, which was occurred naturally, possessed higher thermal stability, mechanical strength, and better barrier properties. Therefore, the composite film prepared by the two matters was not be studied yet.

  1. The film fabrication method is not clear? Add more details.

Replying:

The contents “A certain amount of bentonite powder was dispersed in deionized water by ultrasonication to form a stable suspension. 1 wt % bentonite suspension was prepared by stirring at 1000 rpm for 30 min and then exfoliating by an ultrasonic processor from Scientz-II D(Ningbo Scientz Biotechnology Co., Ltd). This process was repeated three times, followed by centrifugation of the solution at 3800 rpm for10 min. The 1 wt% QH solution was mixed with the obtained1 wt % bentonite suspension, and the volume ratios of the two substances were kept at 1/1. The mixed solution with a total volume of 15 mL was stirred at 30℃for 24 h and then was vacuum filtrated with the filter membrane for 20 min. The obtained film was named as PEI 0%. A series of composite films were obtained by adding different contents of PEI (5%, 10%, 15%, 20%) with the same steps as the PEI 0%. The films obtained with the PEI were named as PEI 5%, PEI 10%, PEI 15%, and PEI 20%, respectively.” have been revised into “5 g bentonite powder was dissolved in 500 mL deionized water to be configured into a 1wt% suspension. 1wt % bentonite suspension was prepared by stirring at 1000 rpm for 30 min, and then expanded and exfoliated by an ultrasonic processor from Scientz-II D (Ningbo Scientz Biotechnology Co., Ltd). This process was repeated three times, followed by centrifugation of the solution at 3800 rpm for10 min. The obtained bentonite suspension was used for composite preparation. The concentrations of QH and PEI were all kept at 1wt%. Hemicellulose-based organic-inorganic films with polyethyleneimine were pre-pared by vacuum suction filtration. Firstly, The QH solution was mixed with the obtained bentonite suspension, and the volume ratios of the two substances were kept at 1/1, obtaining the QH- bentonite matrix. Secondly, different contents of PEI were added (0 wt%, 5wt%, 10wt%, 15wt%, 20wt%) to the QH- bentonite blend liquid. All the mixed solution volume was at 15 mL, and then was magnetically stirred for 24 h at 30 oC. Then, the QH bentonite blend liquid was obtained after vacuum filtration of the filter membrane for 20min. Finally the composite films were vacuum dried for 15min at 80 oC. The film obtained without Pei was designated as PEI 0%. The composite films with different contents of PEI (5wt%, 10wt%, 15wt%, 20wt%) were named as PEI 5%, PEI 10%, PEI 15%, and PEI 20%, respectively.”, and marked in red.

  1. Write the full form once when mentioning for the first instance.

Replying:

Yes, The contents “structure, morphology, optical transparency and thermal stability of the composite films were analyzed by FT-IR, XRD, SEM, TGA” has been revised into “The structure, morphology, optical transparency and thermal stability of the composite films were analyzed by fourier transform infrared spectrometer (FT-IR), X-ray diffraction (XRD), scanning electron microscope (SEM), thermogravimetric analysis (TGA), trans-parency and tensile properties.”, and marked in red.

  1. Improve the quality of all the Figures.

Replying:

All the Figures has been improved as follows, and presented in the manuscript.

  1. Add apparent images of all the fabricated films.

Replying:

The films were all fabricated from the same component with different ratios. Therefore, the shapes and images of the films were little difference. The only image presented in the text was just for typical examples.

  1. Why is only one SEM image is showed in Fig. 2? Add SEM images of all the fabricated composite films.

Replying:

The SEM images of PEI 0% has been added in Figure 2.

Figure 2. SEM images of the surface (A1, B1) and cross section (A2, B2) of the composite films.

  1. Why the residual char content is very high, need explanation?

Replying:

The contents “The unique octahedral structure confers good thermal resistance to bentonite, which is lost a small part of its weight. This has been well demonstrated in studies by Costafreda and Alther.” have been added in Section 3.5, and marked in red.

The Figure 5 and Table 2 have been revised as follows.

Sample

Tonset / oC

T1 / oC

Residual mass / %

PEI 0%

184.7

277.5

55.6

PEI 10%

220.7

279.2

57.3

QH

119.6

273.9

17.3

bentonite

-

-

95.3

Figure 5. TGA curves of QH, bentonite, PEI 0% and PEI 10% (A), the DTA curves of QH (B), he DTA curves of PEI 0% (C), he DTA curves of PEI 10% (D).

Table 2. TGA and DTG data of composite films, bentonite, and QH.

  1. What about the hydrodynamic properties such barrier, hydrophobicity and water solubility of the film? Add this information.

Replying:

Thanks for your good suggestion. The mainly purpose of this study was the fabrication of the composite films, and the simple characterizations were analyzed. The application of the composite was in packaging applications, and the relative properties would be detected for the next step.

  1. Add statistical analysis with significant differences in Table 1 & 2.

Replying:

  Yes, the standard deviation has been added in Table 1.

Table 1. The results of tensile tests of the composite films.

Sample

Thickness (um)

Tensile strength

(MPa)

Tensile strain at fracture(%)

Young’s modulus

(GPa)

PEI 0%

71±0.87

22.67±0.65

0.92±0.22

2.41±0.29

PEI 5%

73±0.93

25.94±0.71

0.51±0.12

4.31±0.59

PEI 10%

76±0.62

28.96±0.42

0.81±0.31

6.08±0.13

PEI 15%

78±0.53

45.54±0.53

0.70±0.31

7.15±0.17

PEI 20%

80±0.24

80.52±0.33

1.30±0.84

8.14±0.04

  1. Authors are advised to add more discussion of the obtained results with up-to-date citations.

Replying:

  Thanks for your good suggestions. The relative contens and new citations have been added in the manuscript.

  1. Also, carefully revise the typos and linguistic errors to make the manuscript error-free.

Replying:

  Yes, all the typos and linguistic writings have been rechecked and revised.

Round 2

Reviewer 1 Report

After corrections the manuscript reads well. I suggest publication.

Author Response

Thank you for your accept.

Reviewer 2 Report

The revised version of the manuscript has improved. The manuscript can be accepted after minor revision.

The authors should add the significant difference of the data in Table 2 using statistical analysis such as SPPS.

Authors should add the image off all the apparent films.

Author Response

  1. The authors should add the significant difference of the data in Table 2 using statistical analysis such as SPPS.

Replying:

Yes, the significant difference analysis has been added in Table 2.

Table 2. TGA and DTG data of composite films, bentonite, and QH.

Sample

Tonset / oC

T1 / oC

Residual mass / %

PEI 0%

184.7±0.97

277.5±0.42

55.6±0.57

PEI 10%

220.7±0.77

279.2±0.39

57.3±0.54

QH

119.6±0.52

273.9±0.48

17.3±0.27

bentonite

-

-

95.3±0.13

  1. Authors should add the image off all the apparent films.

Replying:

The images off all the apparent films has been added in Figure 5.

Figure 5. The images of the composite films (a, b, c, d, e were PEI 0%, PEI 5%, PEI 10%, PEI 15%, and PEI 20%).